# Application of DFT Calculations in Designing Polymer-Based Drug Delivery Systems: An Overview

**DOI:** 10.3390/pharmaceutics14091972

**Published:** 2022-09-19

**Authors:** Oluwasegun Chijioke Adekoya, Gbolahan Joseph Adekoya, Emmanuel Rotimi Sadiku, Yskandar Hamam, Suprakas Sinha Ray

**Affiliations:** 1Department of Chemical, Metallurgical and Materials Engineering, Faculty of Engineering and the Built Environment, Institute of NanoEngineering Research (INER), Tshwane University of Technology, Pretoria 0183, South Africa; 2Department of Electrical Engineering, French South African Institute of Technology (F’SATI), Tshwane University of Technology, Pretoria 0001, South Africa; 3École Supérieure d’Ingénieurs en Électrotechnique et Électronique, Cité Descartes, 2 Boulevard Blaise Pascal, Noisy-le-Grand, 93160 Paris, France; 4Centre for Nanostructures and Advanced Materials, DSI-CSIR Nanotechnology Innovation Centre, Council for Scientific and Industrial Research, CSIR, Pretoria 0001, South Africa; 5Department of Chemical Sciences, University of Johannesburg, Doornforntein, Johannesburg 2028, South Africa

**Keywords:** drug delivery system, DFT, polymer, nanocomposites, quantum mechanics

## Abstract

Drug delivery systems transfer medications to target locations throughout the body. These systems are often made up of biodegradable and bioabsorbable polymers acting as delivery components. The introduction of density functional theory (DFT) has tremendously aided the application of computational material science in the design and development of drug delivery materials. The use of DFT and other computational approaches avoids time-consuming empirical processes. Therefore, this review explored how the DFT computation may be utilized to explain some of the features of polymer-based drug delivery systems. First, we went through the key aspects of DFT and provided some context. Then we looked at the essential characteristics of a polymer-based drug delivery system that DFT simulations could predict. We observed that the Gaussian software had been extensively employed by researchers, particularly with the B3LYP functional and 6-31G(d, p) basic sets for polymer-based drug delivery systems. However, to give researchers a choice of basis set for modelling complicated organic systems, such as polymer–drug complexes, we then offered possible resources and presented the future trend.

## 1. Introduction

One of the most exciting areas of biomedical research is the investigation of innovative drug delivery methods. To improve medication cell/tissue selectivity, rate of release, therapeutic index, and bioavailability, multidisciplinary scientific techniques integrating traditional or engineered technologies are applied. The building elements of these systems are typically biodegradable and bioabsorbable polymers, with their copolymers serving as delivery components [1,2]. To boost the therapeutic efficiency of medications, therapeutic agents, and vaccines, drug delivery systems are used to transport them to a specific location within the body with a regulated release [3]. Computational methodologies, such as DFT, have been developed to bypass time-consuming empirical procedures for the optimization of these formulations.

Density functional theory (DFT) computations, in particular, offer outstanding levels of accuracy with comparable computation time and are more inexpensive in terms of computational resources than other ab initio approaches currently in use. Additionally, it avoids the many electron wavefunction in favor of electron density, and has the potential benefit of dealing with only one function of a single spatial coordinate. Moreover, it employs generalized gradient approximations (GGAs), which use the density gradient to generate a more precise function [4,5,6].

DFT are a strong and low-cost method for revealing a material’s fundamental information, including energy, geometric structure, electrical, and optical characteristics. It offers important theoretical predictions and assistance from the standpoint of material design. It provides crucial information at the levels of atoms, molecules, and unit cells from the perspective of interpreting the results. The influence of element doping on the geometric and electrical characteristics of polymer-based drug carriers, as well as the interaction between the drugs and the nanocarriers, is considerably aided by DFT calculations [7,8].

For example, Kazemi and colleagues used DFT calculations to examine the hydrogen bonding interactions between letrozole and methacrylic acid-trimethylolpropane trimethacrylate copolymers as drug delivery systems [9]. Meanwhile, Karatars and co-workers investigated the interaction of curcumin with a poly(lactic-co-glycolic acid) and montmorillonite composite in a drug delivery system from the first principle calculation [10]. The drug delivery potential of hexagonal boron nitride (h-BN) and PEGylated h-BN (PEG-h-BN) for the delivery of the anticancer medication Gemcitabine (Gem) was investigated by Farzaneh and colleagues using DFT simulations. With an adsorption energy of -15.08 kJ/mol, the drug physically adsorbs into the h-BN surface by the growth of π-π stacking. The findings also show that the PEG group grafting to h-BN resulted in π-π stacking, which is enhanced by the generation of strong hydrogen bonds and resulted in a 20% increase in adsorption energy (−90.74 kJ/mol) [11].

With numerous research investigations carried out in recent years, the utilization of computational material science, in particular, DFT calculations for polymeric DDS, symbolizes a new paradigm in the area of polymer science. However, there have not been any thorough evaluations of this significant topic, though. Therefore, in this work, we provided an evaluation of DFT calculations for predicting the parameters of polymer-based drug delivery systems. First, we outline the most important aspects of DFT while also providing some background information. Then, we looked at the fundamental traits that DFT calculations may predict. As a result, electronic, energetic, thermodynamic, and adsorption properties such as binding energy (Eb) were explored. Moreover, quantum molecular descriptors, including band gap energy (Eg), chemical hardness (η), and electrophilicity (ω), were considered to explain the reactivity of polymer–drug systems. Additionally, to gain insight into the inter- and intramolecular interactions of polymer–drug complexes, natural bond orbital (NBO), atoms-in-molecule (AIM), charge decomposition analysis (CDA), and density of state (DOS) spectra are taken into account. Noncovalent interactions (NCI) and reduced gradient of density (RDG) assays for examining the noncovalent contact between the drug and the nanocarriers were detailed, along with an in-depth explanation of the release mechanism of pharmaceuticals from polymer-based carriers. To conclude this study, we presented a future trend for the use of DFT for drug delivery utilizing polymers and gave insight into computational resources that are essential.

## 2. DFT: A Quick Overview

The first-principles approach is based on quantum mechanics (QM), which expresses the conduction of electrons and atomic nuclei in every condition. The basic equation in this computation is the Schrödinger equation (Equation (1)). The many-body problem occurs in many-electron systems when electrons interact with one another [12,13,14].
(1)ĤΨ=EΨ
where Ψ is the wavefunction, Ĥ is Hamiltonian, and E is the system’s energy.

Several approximations, such as the Born–Oppenheimer approximation, were devised to solve the complex Schrödinger equation. Two theorems proposed by Hohenberg and Kohn served as the foundations of DFT. The Hohenburg–Kohn equation may be written as Equation (2) according to the first theorem.
(2)$$E_{0}=E_{v}\left[\rho_{0}\right]=\bar{T}\left[\rho_{0}\right]+{\bar{v}}_{ee}\left[\rho_{0}\right]+\int \rho_{0}\left(ɍ\right)v\left(dɍ\right)dɍ$$
where $\bar{T}$ is the sum of electronic kinetic energy, ν(ɍ) is the function for nuclear potential energy for an electron at a point ɍ, $\bar{v}$ is electron–electron repulsion, and the overbars denote the average variables.

The second hypothesis proposes a density minimum principle, stating that the ground state energy of any trial electron density $(\rho_{tr}$) cannot be lower than that of the true ground system and may be represented by Equation (3).
(3)$$E_{v}\left[\rho_{tr}\right]=E_{v}\geq E_{0}=\left[\rho_{0}\right]$$

The electronic density *n*(ɍ) is defined by fulfilling the requirement that *n*(ɍ) d ɍ is the probability of finding any electron in the volume d^3^ɍ around ɍ. It is simply |ϕ(ɍ)|^2^ for a single electron with wavefunction ϕ(ɍ). In DFT, we represent the ground-state energy in terms of *n*(ɍ) instead of *Ψ* [15,16].
(4)$$F\left[n\right]=\min_{\Psi \to n}\langle \Psi |\{\bar{T}+{\bar{\nu }}_{ee\;}\}|\Psi \rangle$$

Moreover, the Kohn–Sham technique is a method for calculating atom and molecule energy, structure, and characteristics [17,18].
(5)$$E_{v}\left[\rho \right]=\int \rho \left(ɍ\right)v\left(dɍ\right)dɍ+{\bar{T}}_{s}\left[\rho \right]+\frac{1}{2}\iint \frac{\rho \left({ɍ}_{1}\right)\rho \left({ɍ}_{2}\right)}{{ɍ}_{12}}d{ɍ}_{1}d{ɍ}_{2}+E_{XC}\left[\rho \right]$$

$E_{XC}$ stands for exchange-correlation (XC) energy, which consists of correlation, exchange, columbic correlation, and self-interaction correction. To determine the electronic structural characteristics of polymer-based drug delivery systems, the DFT method has been widely employed. The traditional DFT approach, which uses LDA and GGA exchange-correlation (XC) functionals, produces a band gap that is underestimated. Several approximations such as DFT+U, DMFT, hybrid functionals (B3LYP, PBE0, and HSE), GW approximation, etc., have been proposed to attain better results. Although hybrid functionals and GW give a more exact value of band gap, they come at a hefty computational expense.

The materials science community has access to several DFT codes. The programs are based on various potentials, basis sets, exchange-correlation functionals, and Schrödinger equation-solving techniques. ABINIT, ADF, CASTEP, DMol^3^, ONETEP, Gaussian, and GAMESS are some examples of DFT programs. Table 1 summarizes the different DFT-based software, the functionals, and the basis set used in polymer-based drug delivery systems.

## 3. Fundamental Properties of Polymer-Based Drug Delivery Systems

DFT computations may be used to derive several characteristics of polymer-based drug delivery systems [11]. To obtain the desired feature under examination, researchers might use a variety of strategies. However, Figure 1 presents a common procedure for applying DFT calculations. The creation of the drug’s and excipient polymer’s molecular structures often marks the beginning of the procedure [38,39]. Additionally, these structures may be imported from both public and commercial molecular structure databases, including the Protein Data Bank [40], PubChem Structure [41,42], and databases for polymer materials. The energy of each particular molecular structure can then be reduced by performing an initial structural relaxation. This enables the drug’s binding energy to be computed when an adsorption calculation is run to determine the most thermodynamically advantageous adsorption sites. On the other hand, interaction energy will be calculated when an adsorption calculation is performed without first optimizing the structure. Ground state calculations are used to further relax the drug–polymer combination that has formed. The generated complex may then be utilized to forecast excited state attributes, such as UV, IR, and using frequency calculations. To examine the thermodynamic properties, the optimized complex’s total energy may be determined. The energy gap and molecular orbital characteristics of the drug–polymer configurations can be predicted using electronic simulations in the interim [43]. The many DFT computed characteristics of drug delivery systems based on polymers are explored in the following paragraphs.

### 3.1. Optimized Geometry and Adsorption Properties

One way to investigate the interaction of drug molecules with the polymer-based carrier is to calculate binding energy from the first principle. Both an extremely strong and a feebly weak connection are detrimental to delivering pharmaceuticals and therapeutic substances. One objective of carrier design is to achieve the best possible balance between payload protection and payload release, maximizing transfection [34]. There are several techniques and DFT codes that can be used to model the adsorption of drug molecules onto the polymeric carrier. The drug can be docked on the polymer carrier with a module such as a Glide module in Schrodinger [19]. Alternatively, to determine the adsorption site, the Monte Carlo technique can be employed through the Adsorption Locator module [44] in the material studio to sample various configurational spaces to predict the most stable and optimized binding location of the drug on the carrier. Thereafter, the total energy of the drug–carrier complex, the drug, and the carrier is calculated by performing geometry optimization calculations. The geometry optimization is set to increase accuracy and ease of application by using structural analysis to find the most stable arrangement [45,46,47,48].

When simulating macromolecules such as polymer, van der Waals (vdW) interaction is often accounted for, as such, long-range dispersion correction terms (DFT-D), e.g., Grimme’s D3 [49,50], Grimme’s D4 [51,52], and Tkatchenko–Scheffler (TS) [53,54], are used. Moreover, basis set superposition error ($\delta_{\mathrm{BSSE}}$) is employed to correct binding energies, that are obtained by the Boys–Bernardi counterpoise, while zero-point vibrational energies (ZPVE) are considered in reactive systems [11,55,56].

The dispersion corrected adsorption energies ($E_{\mathrm{ads}}$) of interacting components in the investigated complex may be estimated using the following Equations (6) and (7):(6)$$E_{\mathrm{ads}}=E_{\mathrm{drug}-\mathrm{carrier}\;\mathrm{complex}}-\left(E_{\mathrm{drug}}\;+E_{\mathrm{carrier}}\right)$$
(7)$$E_{DFT-D}=E_{DFT}+E_{Dp}$$
where $E_{\mathrm{drug}}$, $E_{\mathrm{carrier}}$, and $E_{\mathrm{drug}-\mathrm{carrier}\;\mathrm{complex}}$ are the energies of drug, carrier, and complex, respectively. $E_{Dp}$ is the long-range dispersion correction or basis set superposition error (BSSE), which may be adjusted for all interaction energies using the counterpoise correction approach.

The calculated binding energies of drug–carrier complexes are usually negative in drug delivery systems, indicating an exothermic reaction [57,58,59,60,61]. According to Masoumi and colleagues, on the DFT investigation of the electronic properties of SiO_2_ nanotube (SiO_2_NT) interacting with a copolymer poly lactic-co-glycolic acid (PLGA). It was reported that the adsorption energy between the PLGA and SiO_2_NT is greater than the carbon nanotube (CNT) after geometry optimizations were carried out. Because of this, the exothermic reaction between PLGA and SiO_2_NT is about an order of magnitude larger than the reaction between PLGA and CNT [62]. 

Similarly, Karataş and colleagues used the B97-D functional with the TZVP basis set in a DFT investigation on the interaction of curcumin in a drug delivery system containing a composite with montmorillonite and poly(lactic-co-glycolic acid). The interaction between two lactic acid (LA) and glycolic acid (GA), monomers, and two PLGA copolymers containing five units of LA and GA molecules was demonstrated, as illustrated in Figure 2. The shorter PLGA copolymer dimer model is stabilized by both intra-molecular (1.90 Å) and inter-molecular (1.84 Å) hydrogen bonding (Figure 2). This reveals that individual acid monomer interactions are preferable to those in the block copolymer dimer [10]. Since the geometrical parameters may not be adequate for a thorough examination of the analyzed intermolecular interactions at the studied configurations, other interaction properties are investigated.

### 3.2. Electronic Properties and Quantum Molecular Descriptors

The highest occupied molecular orbital (HOMO), the lowest unoccupied molecular orbital (LUMO), and the HOMO-LUMO energy gaps (Eg) are strong indicators of the electronic properties of an interaction system. The HOMO and LUMO are also known as the frontier orbitals and are crucial variables that provide qualitative information on the excitation characteristics of modelled substances. Meanwhile, the energy gap (Eg) is a valuable tool for determining the chemical reactivity of interreacting molecules in a drug delivery system [63,64,65].

As illustrated in Figure 3, Cortes et al. presented the frontier molecular orbital (the HOMO and LUMO) knowledge of antimalarial drug (chloroquine, primaquine, and amodiaquine) interactions using the acrylamide-case hydrogels model [22]. It is worth noting that HOMO orbitals are positioned in a specific molecular location for the antimalarial drugs, which indicates a high probability of better interaction between the drugs and hydrogel as the monomer chain increases.

The quantum molecular descriptors in DFT calculation are the chemical-reactivity descriptors (μ, ƞ, s, ω, ∆N) utilized to explain the interplay between polymer-based excipients and drug activity in drug delivery systems. These descriptors are generally the chemical potential (μ), the global hardness (η), the softness (s), the electrophilicity index (ω), and the maximum charge transfer index (∆N). The global hardness or a molecule’s hardness (η) is specifically related to its resistance to changes in electrical distribution and has shown to be effective in the rationalization of chemical reactions, whereas the molecule’s softness is the inverse (s = 1/η) of the global hardness. The electronegativity (χ) of an element describes the degree to which it tends to acquire electrons and generate negative ions in a chemical process of the delivery system. The electrophilicity index (ω) estimates the reactivity of a molecule by assessing its capacity to accept electrons, whereas the chemical potential (μ) shows the direction of flow of electrons from the higher μ to the lower μ until the chemical potential becomes equilibrated.
(8)$$ECT=\left({\left(\Delta N_{\max}\right)}_{Drug}-{\left(\Delta N_{\max}\right)}_{Excipient}\right)$$
(9)$$\Delta N_{\max}=-\mu /\eta$$
(10)$$\Delta N_{\max}=2\omega /\chi$$
(11)$$\chi =-\left(E_{HOMO}+E_{LUMO}\right)/2$$
(12)$$\eta =\left(-E_{HOMO}-(-E_{LUMO}\right))/2$$
(13)$$\mu =\left(E_{HOMO}+E_{LUMO}\right)/2$$
(14)$$\omega =\chi^{2}/2\eta$$
(15)$$\omega =\mu^{2}/2\eta =\left(-\frac{\mu }{\eta }\right)\chi \Delta N_{\max}/2$$

Equations (8)–(15) show the mathematical link between the electronic properties and quantum molecular descriptors. In an adsorption process of a delivery system, η is calculated to be half of the HOMO-LUMO energy differences, μ is defined as the average of HOMO and LUMO energies, and ω can be calculated according to Equations (14) and (15).

The softness, hardness, electrophilic index, and electronegativity of the targeted area and drug can be connected to the reaction rate, structural stability, polarizability, and molecular toxicity of any chemical compounds in the biochemical system [66,67]. For instance, if the energy gap (band gap), i.e., the difference between the HOMO and LUMO of any molecule is small, consequently, the molecule is highly polarizable, and often exhibits strong chemical reactivity with low stability. It should be noted that a good-electrophilic molecule has a higher value of both μ and ω, compared to a good-nucleophilic molecule [68]. Therefore, the chemical descriptors must be computed using DFT to investigate the electronic properties of the drug on the target cell throughout the polymer-based drug delivery process.

### 3.3. Thermodynamics Properties

The density functional theory (DFT) is applied to drug delivery systems to provide a deeper understanding of essential energy changes due to adsorption and the energetic magnitudes of molecular-level insights to explain the thermodynamic behavior of the interacting system. The thermodynamic parameters such as enthalpy (ΔH), entropy (ΔS), and Gibbs free energy (ΔG), for drug systems, are determined at temperature (298 K) and pressure (1 atmosphere) [22,69,70]. The negative values of ΔH imply exothermic reactions, whereas the positive values of ΔG suggest non-spontaneous reactions. Positive entropy indicates an increase in the disorder of the delivery system ΔS > 0 [43,71]. 

The Gibbs free energy function ΔG, of the drug delivery system at a constant temperature could be obtained by adding the value of enthalpy to the entropy-temperature product as expressed in Equation (16):(16)ΔG=ΔH−TΔS

In a recent DFT study using B3LYP-D/6–31 + G (d, p), Alireza et al. reported an improvement in the drug delivery efficacy and the anti-inflammatory activity of sulfasalazine in two configurations with PLGA in water and dichloromethane environments. The optimized configurations of the model (complex A and B) are shown in Figure 4. In complex A, the sulfasalazine drug establishes a double hydrogen bond with the PLGA microparticle via its carboxylate group, thereby making it more stable than complex B. 

In the thermodynamic approach, the Gibbs free solvation energy (ΔG_solv_) for complex A and B was reported to be − 0.15 and + 0.14 eV, respectively, while their enthalpy (ΔH_solv_) values were − 0.65 and − 0.38 eV, respectively. The negative values of ΔG_solv_ and ΔH_solv_ in complex A indicate the spontaneous contact and exothermic process between the sulfasalazine and the PLGA interactions. The difference in values of ΔG_solv_ demonstrates the entropic effects between the adsorption process of the drug and carrier in the two complexes [72].

### 3.4. Release Mechanism of Drugs from Polymer-Based Carriers

Drug molecules are adsorbed on the excipient with appropriate strength and driven to the target location, where they are released, into a biological system. To model the release profile of the drug/carrier delivery system, geometry optimizations are performed for the drug/carrier complexes with additional hydrogen proton (protonation process) to model the lower pH environment. Drug release from the carrier surface is very important and can be achieved through external or internal stimuli [73]. The proton attack in the delivery mechanism causes drug molecules to separate from the carrier surface. The released drug can thereby bind with the target site to treat the disease effectively [67]. The recovery time for drug desorption from a carrier can be predicted from the following transition theory:(17)$$\tau =v_{0}^{-1}\;exp\left(\frac{-E_{b}}{KT}\right)$$
where $v_{0}$ represents the attempt frequency (~10^12^ s^−1^), T is the temperature, and k is the Boltzmann’s constant (~1.99 × 10^−3^ kcal/mol·K).

Equation (17) above relates the drug’s adsorption energy on the carrier to the recovery time. Bagheri et al. reported a very short recovery time of about 0.02 s at room temperature for the release of the drug adrucil from the surface of the Si-doped phagraphene. Thus, it can be said that phagraphene is a good option for the transport of the medicine adrucil thanks to Si-doping [74]. Similarly, Xu and colleagues used density functional theory to explore the release mechanism of an anti-cancer medication hydroxyurea (HU) molecule from the adsorbed boron nitride fullerenes (BNFs) surface. The HU drug was, therefore, physically adsorbed on the BNF surface due to the hydrogen bond interaction between the hydroxyl (OH) groups of the HU drug and the nitrogen (N) atoms on the BNF surface, as observed during the adsorption process. Therefore, the possible release mechanism of the drug from the surface of the carrier is a result of the proton attack, as illustrated in Figure 5 [67].

### 3.5. Charge Decomposition Analysis (CDA) and Density of State (DOS) Spectrum

In a drug delivery complex, the charge decomposition analysis (CDA) aids in understanding orbital intrinsic properties and throws light on the mode of charge transfer between fragments of molecular orbitals, as well as the charge transfer transition in its isolated state for achieving charge equilibrium. The density of states (DOS) analysis determines the number of states in a unit energy interval of a system; it is the most important metric for analyzing fragment interactions and determining their molecular orbital (MO) energy level contributions in the complex. The CDA and DOS spectra both offer a visual representation of the interacting components [75,76,77,78]. Multiwfn is a multifunctional software designed to make visual exploration of electron and molecule structures in wavefunction analysis easier [79].

By applying DFT calculations at the B3LYP/6-31G(d) level, Kazemi and coworkers studied the hydrogen bonding interactions between methacrylic acid-trimethylolpropane trimethacrylate copolymers and letrozole. The DOS spectra between drug 1 and copolymer 4 in compounds **7**, **8**, and **13** provide insight into the nature of intramolecular interactions. Figure 6 shows the display of the hybridization of the drug and the copolymer. As a result, the DOS spectra are used to quantitatively determine the existence of interactions [9].

According to Mehvish and coworkers in their DFT approach to the therapeutic potential of graphitic carbon nitride as a drug delivery method for cisplatin. The incorporation of additional MO energy levels in the g-C3N4–cisplatin complex resulted in a larger value of HOMO (5.56 eV) and a lower value of LUMO (3.42 eV), which led to the reduction of Eg (band-gap). This narrower energy gap (Eg) between the border MOs aids charge transport in the g-C3N4–cisplatin complex. These findings show that the cisplatin medication was successfully attached to the g-C3N4 molecule. As seen in the orbital-interaction diagram CDA-diagram in Figure 7, charge transfer happens as a result of new molecular energy levels being introduced into the complex’s MOs by the medication and carrier components. These findings show that the cisplatin medication was successfully attached to the g-C3N4 molecule. The CDA and DOS data both demonstrated how the inclusion of additional MOs in the g-C3N4–cisplatin complex as a result of the intermixing of cisplatin and g-C3N4 Mos contributed to a substantial drop in the energy gap (Eg = 2.15 eV) and its subsequent participation in charge transfer between two fragments [78,80].

### 3.6. Natural Bond Orbital (NBO) and Atoms-in-Molecule (AIM) Analysis

The electron charge-transfer mechanism is an important component in the adsorption of an adsorbate on an adsorbent. While natural bond orbital (NBO) analysis is effective for examining intermolecular charge transfer with intramolecular and intermolecular interactions between bonds in a molecular system, AIM analysis is a more efficient method for determining the nature of intermolecular interactions [11]. Furthermore, NBO calculations in the delivery system can be utilized to determine the direction and quantity of charge transfer from one unit to the next in the equilibrium geometries during the adsorption process [81].

According to the NBO and AIM analysis performed by Bazyari-Delavar and colleagues on interactions between polyester dendrimers and ibuprofen using DFT, the stability of the drug–carrier complex could be attributed to the intermolecular hydrogen bonds formed between the functional groups of the interacting components, i.e., the carriers (polyester G1 dendrimer) and drug (ibuprofen) molecules. To further understand the encapsulation mechanism, the G1: Ibu complex is stabilized by hydrogen bonds produced between the hydroxyl groups of dendrimer and the carboxylate groups of Ibupofen [31]. 

### 3.7. Noncovalent Interactions (NCI) and Reduced Gradient of Density (RDG) Analyses

Since the non-covalent interactions (NCI) between drug and carrier play an important role in drug offloading at the target location in the drug delivery system. The reduced density gradient (RDG) analysis is used to investigate the various non-covalent interactions that exist inside molecules [38,78]. The NCI-RDG analysis is utilized to learn more about the intermolecular interactions, repulsive interactions, and nonlocalized dispersion among the reacting components. This method has emerged as a useful tool and efficient method used to visualize weak interactions such as hydrogen bonding, van der Waals (vdW), and strong repulsive (steric) interactions within the non-covalently interacting structures.

In general, the electron densities of vdW interactions are relatively low, whereas the electron densities of interactions due to the hydrogen bond and the steric effect are very high. Olfa and colleagues revealed the distinct forms of non-covalent interactions in the antiviral activities of hybrid hydroxychloroquine in the treatment of coronavirus 2 (SARS-CoV-2) investigations using the DFT method at the B3LYP/6-31G* level of theory. Figure 8 depicts the zones of interaction between hydroxychloroquine and hydroxychloroquine sulfate. There are three areas of contact, as can be shown. vdW interactions are shown by the green zones. Repulsive interactions (steric effects) are mostly confined at the cycle level, as indicated by the red color. The blue patches indicate the presence of hydrogen bonding. These interactions are based on electron density characteristics [82].

Farzad and coworkers also reported distinct forms of non-covalent interactions based on the color-filled RDG isosurface during the investigation of the probing impact of polyethylene glycol (PEG) on the adsorption mechanisms of Gemcitabine (Gem) drug on the hexagonal boron nitride (h-BN). The RDG vs. sign(λ_2_)ρ and RDG-based NCI isosurface plots of two configurations of the drug–carrier complex were predicted by the Multiwfn program as shown in Figure 9. Because of the formation of π-π stacking between the drug and the carrier, the green region in the first configuration is more than the green region in the functionalized complex, as shown in Figure 9a,b, respectively. While in Figure 9b, in the second configuration, the formation of the hydrogen bonds causes the blue area to be increased [11].

### 3.8. Molecular Electrostatic Potential (MEP) Diagram

The molecular electrostatic potential (MEP) diagram is a useful property to study reactivity to electrophilic or nucleophilic attacks and visualize the location of the electron density in the drug delivery systems. In the establishment of a hydrogen bond, there are two separate areas, such as positive and negative, that represent the H-donor and H-acceptor features of molecules, respectively. The map of MEP is color-coded. The positive region is depicted in blue for a nucleophilic attack, whereas the negative region is indicated in red for an electrophilic attack and green indicates zero potential areas in the molecule [83,84].

The use of MEP to investigate the interaction between the molecules of chitosan and the gene carrier (nucleobase) was effectively reported in a DFT research by Deka et al. Chitosan interacts with nucleophilic sites in nucleobases during the formation of the chitosan–nucleobase adduct by forming hydrogen bonds between the amino-H atom of chitosan and the O or N atom of nucleobases. Figure 10 shows an isosurface diagram of chitosan and guanine, with a negative charge area surrounding the N7 and O atoms of guanine serving as H-acceptors and a positive charge density around the H atoms of the NH_3_ group acting as H donors [34].

The electron density of drug–excipient complexes may also be shown using electrostatic potential (ESP) charting inside isosurface plots in conjunction with MEP maps. In a DFT study by Singh et al. [83] on the vibrational spectra of 5-chlorouracil with molecular structure, using DFT/Gaussian 09 with GAR2PED code. The MEP surface plots were traced using different colors to represent the electrostatic importance together with iso-surface contours electron density EPS mapping of 5-chlorouracil as shown in Figure 11a,b, respectively. These plots are the basis for the understanding and visualization of the relative polarity of the molecule [59,85].

On surface mapping/arrays, MEP increases in the following order: red, orange, yellow, green, and blue. In this case, negative MEP reflects the attraction of a proton in the red region owing to the concentrated ED in the molecule, whereas positive MEP indicates the repulsion of a proton in the blue zone due to the atomic nuclei regions and the low concentration of ED in the molecule. The red hue represents the negative area and suggests nucleophilic reactivity, whereas the blue color depicts electrophilic reactivity in the positive region of MEP surface plots.

## 4. Conclusions and Outlook

In this work, we have explored different fundamental properties that are used in the calculation and prediction of drug delivery performance in the biological system. As mentioned in this article, we highlighted how DFT has been widely utilized to investigate the interaction, thermodynamic, electronic, charge transfer, and release mechanism aspects of polymer-based drug delivery systems. 

Despite the use of various DFT software to solve drug delivery behavior, the Gaussian software has been widely used, particularly with the B3LYP and 6-31G(d, p) functional and basic sets, respectively. It is envisaged that the logic of rational drug delivery systems and the continuous improvement in computing power of DFT software will spur increased multidisciplinary efforts for the future of nanomedicine from the atomic perspective at a low cost with accurate and reliable quantum mechanic calculations. Therefore, to analyze the drug’s absorption and release on the target cell throughout the drug delivery process, the effectiveness of a drug polymer-based carrier in a biological system must be evaluated using DFT. Although, it will need a lot of computational resources to utilize DFT simulations for macromolecular systems such as the polymer–drug complex. In addition, optimizing the geometry of such complex structures may prove highly challenging. To circumvent these problems requires expert knowledge, from choosing the appropriate functional/basis set for computations with long-range dispersion correction to adjusting parameters such as iteration, self-consistent field (SCF), and cutoff energy.

Moving forward, we recommend that researchers perform DFT calculations, MD simulations, and experimental assays to validate the identified drug and its interactions with the target sites for biomedical applications. Software packages such as Gaussian, Orca, Schrodinger Suite, and Materials Studio, among others, offer robust quantum-mechanical code for the simulation of polymer–drug systems. Meanwhile, a library for reading and interacting with basis sets may be found in Basis Set Exchange (BSE) [85], an open-source data repository for basis sets used in quantum chemistry. 

The integration of machine learning (ML) with DFT, although presently in its infancy, can accelerate the discovery and predict the suitability of polymeric nanocarriers for drug delivery. New theories and a step toward the development of effective hybrid nanomaterials might be provided by ML [86]. Useful ML algorithms have been reviewed in our previous work [1] which can pave the way for the development of a robust ML workflow for screening and predicting the properties of the drug–polymer complex.

## Figures and Tables

**Figure 1 pharmaceutics-14-01972-f001:**
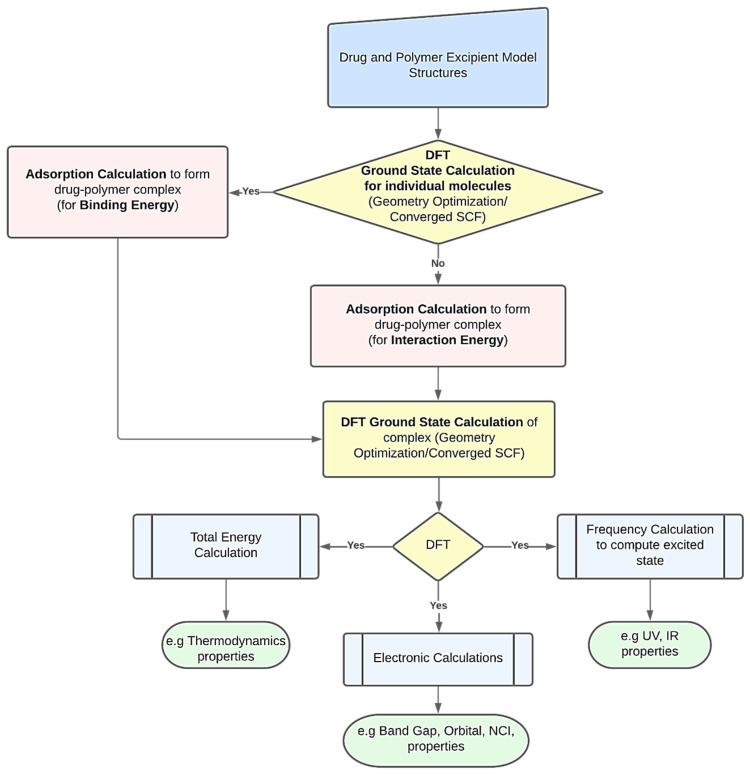
A schematic workflow for employing DFT calculations in polymer-based drug delivery systems.

**Figure 2 pharmaceutics-14-01972-f002:**
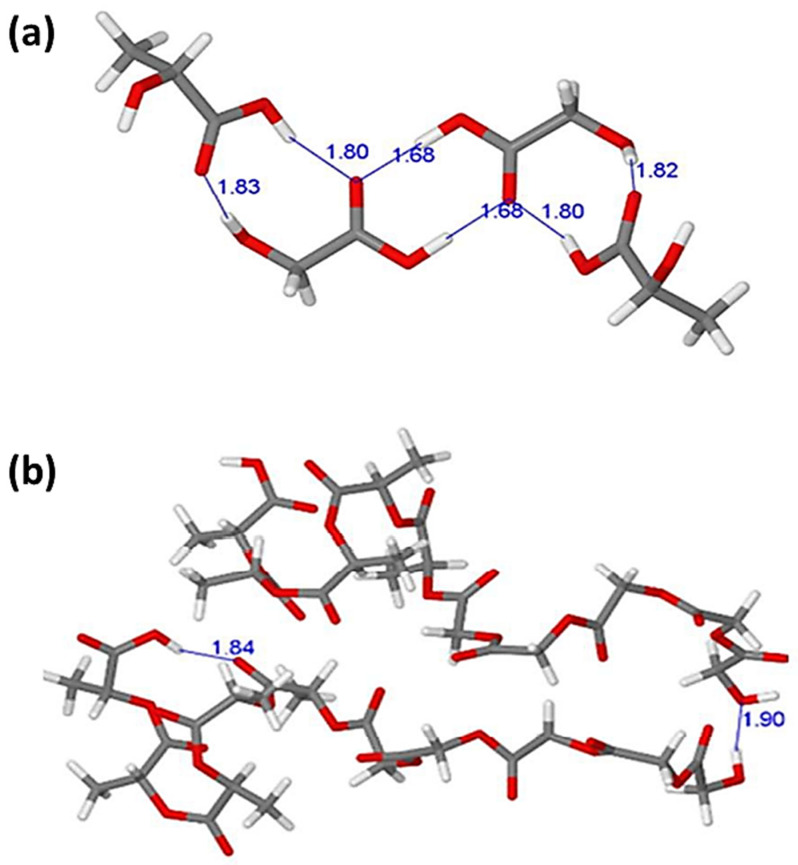
The optimized configurations for the dimer structures of (**a**) the GA-LA-LA-GA tetramer and (**b**) the 5 units LA-5 units GA (PLGA) copolymer dimer. The lengths of the bonds are expressed in angstrom (Å). Reproduced with permission from ref. [10]; Copyright 2022, Royal Society of Chemistry.

**Figure 3 pharmaceutics-14-01972-f003:**
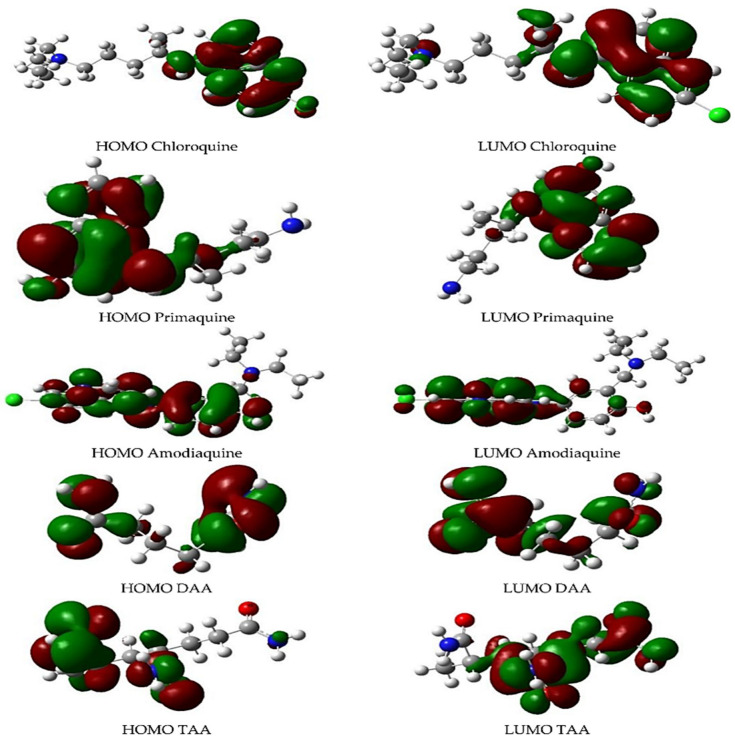
The frontier orbital for the interaction between the antimalarial drugs (chloroquine, primaquine, and amodiaquine) and the acrylamide-base hydrogel. Reproduced from ref. [22]; this is an open-source publication.

**Figure 4 pharmaceutics-14-01972-f004:**
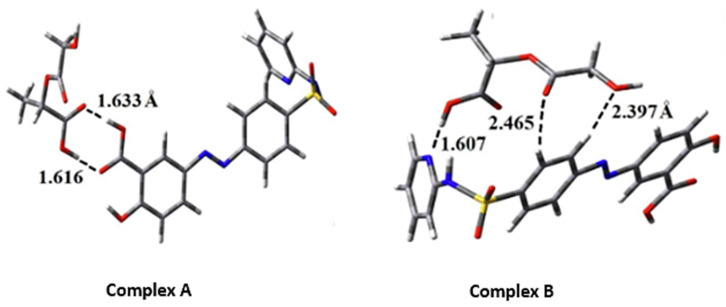
Depicts the two different sulfasalazine and poly(lactic-co-glycolic acid) microparticle configurations. Reproduced with permission from ref. [72]; Copyright 2021, Elsevier Science Ltd.

**Figure 5 pharmaceutics-14-01972-f005:**
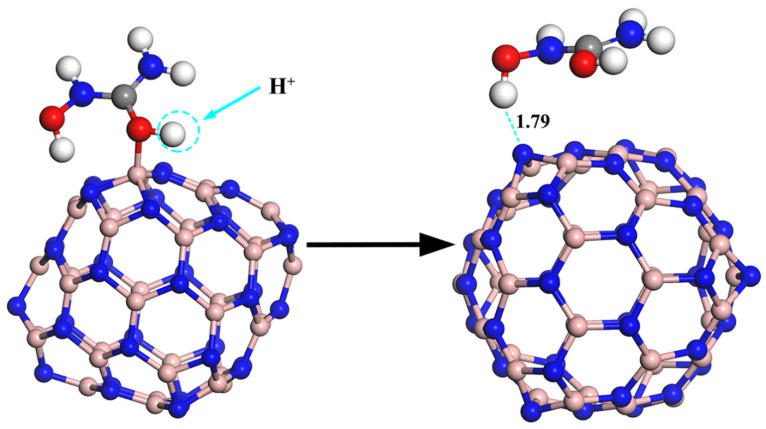
The optimized configuration of the structure of protonated hydroxyurea on the B36N36 surface demonstrates drug release. Reproduced with permission from ref. [67]; Copyright 2020, Elsevier Science Ltd.

**Figure 6 pharmaceutics-14-01972-f006:**
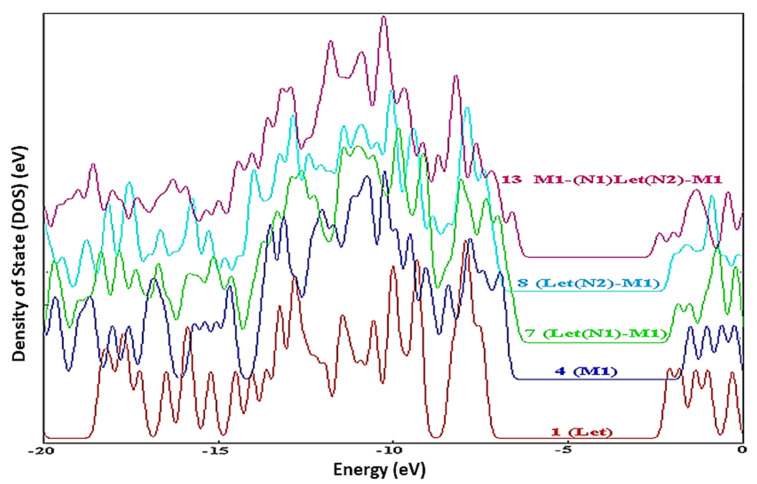
The density of states (DOS) spectra of the drug 1 and copolymer 4 in compounds **7**, **8**, and **13**. Reproduced with permission from ref. [9]; Copyright 2016, World Scientific Publishing Co Pte Ltd.

**Figure 7 pharmaceutics-14-01972-f007:**
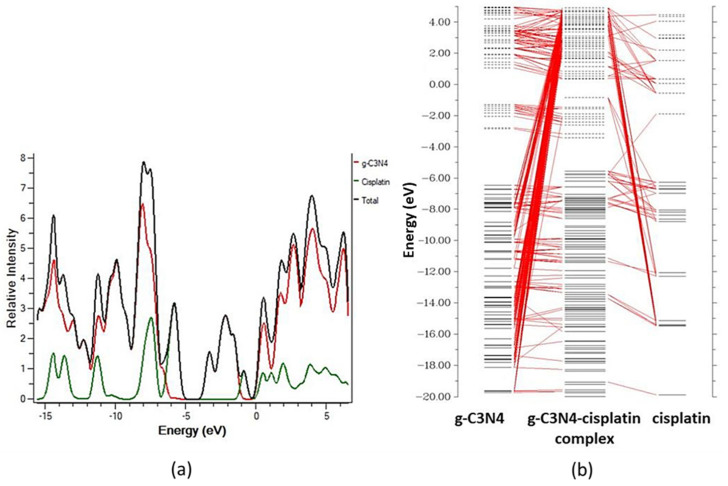
(**a**) g-C3N4, cisplatin, and the g-C3N4–cisplatin complex DOS spectra; (**b**) CDA of the g-C3N4–cisplatin complex. Reproduced with permission from ref. [67]; Copyright 2020, Elsevier Science Ltd.

**Figure 8 pharmaceutics-14-01972-f008:**
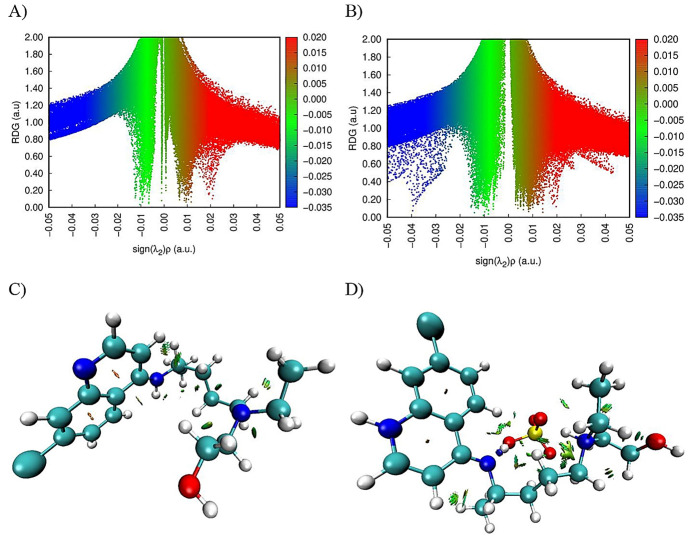
Graphical illustration of the reduced density gradient vs. electron density (**A**,**B**) and the various forms of interactions (**C**,**D**) of hydroxychloroquine and hydroxychloroquine sulfate compounds. Reproduced with permission from ref. [82]; Copyright 2021, Wiley.

**Figure 9 pharmaceutics-14-01972-f009:**
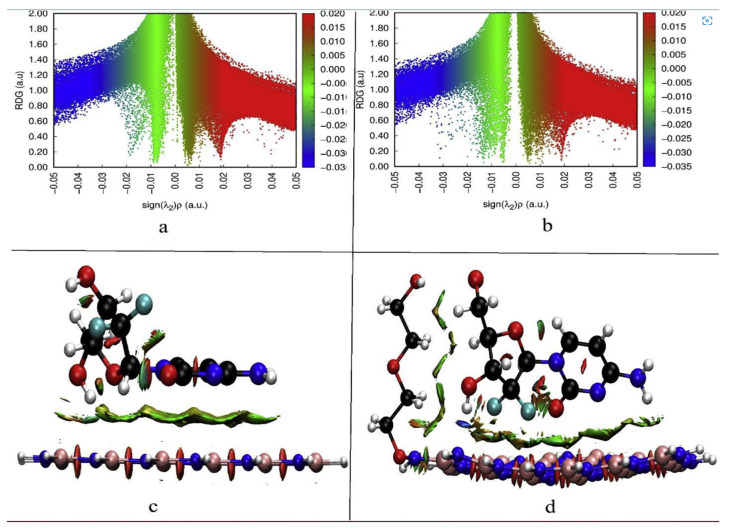
The RDG vs. sign(λ_2_)ρ plots (top) and the color-filled RDG iso-surfaces (bottom) of the drug–carrier complex depict noncovalent interaction (NCI) regions for the non-functionalized (h-BN) (**a**,**b**) and functionalized (PEG-h-BN) (**c,d**) model. Reproduced with permission from ref. [11]; Copyright 2020, Elsevier Science Ltd.

**Figure 10 pharmaceutics-14-01972-f010:**
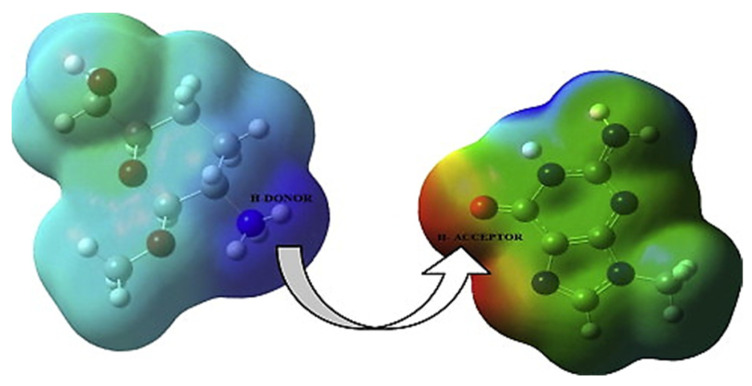
MEP isosurface diagram of the complex model for chitosan and guanine. Reproduced with permission from ref. [34]; Copyright 2015, Elsevier Science Ltd.

**Figure 11 pharmaceutics-14-01972-f011:**
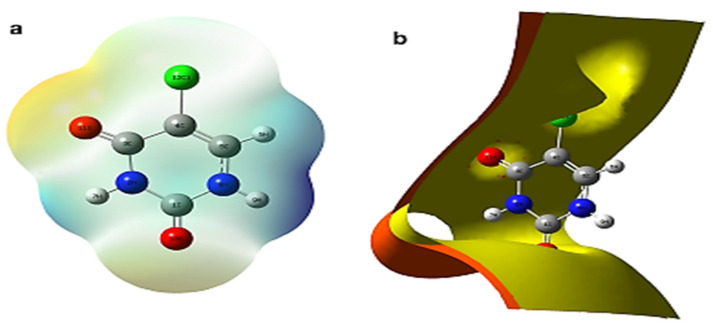
(**a**) The surface plots of MEP and (**b**) the iso-surface contours electron density EPS mapping of 5-chlorouracil. Reproduced with permission from ref. [83]; Copyright 2022, Springer.

**Table 1 pharmaceutics-14-01972-t001:** Calculations in polymer-based drug delivery systems using DFT software applications.

Polymeric Carrier	Drug/Therapeutic Agent	DFT Code	DFT Functional/Basis Set	Ref.
Polylactic-co-glycolic acid and montmorillonite	Curcumin	COSMO module of TURBOMOLE V6.1	B97-D/TZVP	[10]
Disulfur-bridged polyethyleneglycol/DOX nanoparticles	Chlorin e6 (Ce6)	Jaguar module in Schrödinger	M06-2x/6-311(d,p) method	[19]
Hyaluronic acid and Zr-based porphyrinic MOF (HA-PCN-224)	Doxorubicin	DMol^3^ program	PBE/DNP 4.4	[20]
polymer membranes (PDMS, PVA, PUI, PTMG-650-MDI-AP, PPG-725-MDI-AP, PCL-1250-MDI-AP)	Pure liquid solvents	Dmol^3^ module in Accelrys Materials Studio	GGA/VWN-BP/DNP v4.0.0	[21]
PEG-h-BN	Gemcitabine GEM	GaussView software, Gaussian 09 package	M06e2X/ 6-31G(d,p)	[11]
Methacrylic acid-trimethylolpropane trimethacrylate copolymers	Letrozole	Gaussian-98	B3LYP and B3PW91 levels/ 6-31G(d)	[9]
Acrylamide-base hydrogel	Antimalarial (chloroquine, primaquine, and amodiaquine)	-	wb97xd/6-31++G(d,p)	[22]
Chitosan, silicon dioxide, and graphene oxide	Cisplatin	Gaussian 09	B3LYP/LANL2DZ	[23]
Graphene oxide/polyethylene glycol	Sumatriptan	Gaussian	B3LYP/6-31+G(d)	[24]
Polylactic-co-glycolic acid	Anti-cancer sgents of thiazoline	Gaussian 03	B3LYP/6-31G(d)	[25]
Folate functionalized poly(styrene-alt-maleic anhydride), PSMA		Gaussian 09	B3LYP/6-31G(d,p)	[26]
Polyheterocycles	Antischistosomiasis (praziquantel, niclosamide ethanolamine salt, niclosamide, and trichlorfon)	-	B3LYP	[27]
Poly-carboxylic acids functionalized chitosan	Cisplatin	Gaussview 05.	-	[28]
PEGylated fluorinated graphene	Camptothecin (CPT) and doxorubicin (DOX)		-	[29]
Chitosan nanocomposite	Ifosfamide anticancer	COMPASS	-	[30]
Polyester dendrimers	Ibuprofen	Gaussian 09 W	M06-2X/6-31G(d)	[31]
Polyamidoamine (PAMAM) and polyester dendrimers	Favipiravir	-	M06-2X/6-31G(d)	[32]
Hyperbranched polysiloxane containing β-cyclodextrin	Ibuprofen	-	B3LYP/6-31G(d)	[33]
chitosan	Nucleobase (DNA/RNA)	-	B3LYP/6-31++G(d,p)	[34]
Poly(O-vinyl carbamate-alt-sulfones)	Mucosal drug (Rhodamine B)	-	B3LYP, 6-311++G- (d,p)	[35]
PAMAM dendrimers	5-Fluorouracil	Gaussian 09	B3LYP/6-31G(d,p) and M06-2X/6-31G(d,p)	[36]
Chitosan nanoparticle	Hydroxyurea	Gaussian 09	B3LYP/6-31G(d,p) and M06-2X/6-31G(d,p)	[37]
Polyethylene glycol-based nanocomposite	Cephalexin	DMol^3^	B3LYP	[4]

## Data Availability

Not applicable.

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
