# Peer review of "Application of DFT Calculations in Designing Polymer-Based Drug Delivery Systems: An Overview"

_pharmaceutics, 2022, doi:10.3390/pharmaceutics14091972_

Round 1
Reviewer 1 Report
In this review the authors describes the fundamentals and applications of DFT in drug delivery approaches. Although it is a well written review, I suggest the authors to consider the following comments.
1) I suggest the authors to elaborate a little about the fundamental difference between the DFT and traditional methods , as it will provide a basic introduction to the readers about the DFT. Also authors are suggested include the potential draw backs of DFT in the review.
2) I suggest the authors to consider including the recent work by Ye et al., on the COVID-19 drug modeling, Drug Discovery Today (2022) 27: 1411-1419 (doi: 10.1016/j.drudis.2021.12.017).
Author Response
In this review the authors describes the fundamentals and applications of DFT in drug delivery approaches. Although it is a well written review, I suggest the authors to consider the following comments.
1) I suggest the authors to elaborate a little about the fundamental difference between the DFT and traditional methods , as it will provide a basic introduction to the readers about the DFT. Also authors are suggested include the potential draw backs of DFT in the review.
Response: Thank you for your comments. The fundamental difference between the DFT and traditional methods has been added. The potential draw backs of DFT as applied to polymer-drug system has been conclusively included.
2) I suggest the authors to consider including the recent work by Ye et al., on the COVID-19 drug modeling, Drug Discovery Today (2022) 27: 1411-1419 (doi: 10.1016/j.drudis.2021.12.017).
Response: Thank you for your comments, the reference has been included.
Reviewer 2 Report
The review article written by Adekoya et al "Application of DFT Calculations in Designing Polymer-Based Drug Delivery Systems: An Overview " highlight how DFT is an important asset to be used to investigate the interaction, thermodynamic, electronic, charge transfer, and release mechanism aspects of polymer-based drug delivery systems. Overall, this article is well written and organized and fits well within the scope of the journal.
What is the main question addressed by the research? This review highlight how DFT is an important asset to be used to investigate the interaction, thermodynamic, electronic, charge transfer, and release mechanism aspects of polymer-based drug delivery systems
Is it relevant and interesting? Density Functional Theory is progressively becoming vital for the drug designing process. Therefore, I consider an interesting and to date topic
How original is the topic? This is an original piece
What does it add to the subject area compared with other published material? This review camparitive summary of DFT softwares, with emphasizes on the Gaussian software to understand the drug delivery behavior. It features possible resources for the modulation of complicated organic systems, such as polymer-drug complexes
Is the paper well written? Is the text clear and easy to read? well written and organized
Are the conclusions consistent with the evidence and arguments presented? YES
Do they address the main question posed? YES
Author Response
Response: Thank you for your comments.